# Research on Relationships between Sexual Identity, Adverse Childhood Experiences and Non-Suicidal Self-Injury among Rural High School Students in Less Developed Areas of China

**DOI:** 10.3390/ijerph16173158

**Published:** 2019-08-29

**Authors:** Xuyang Li, Huilie Zheng, Winter Tucker, Wenyan Xu, Xiaotong Wen, Yixiang Lin, Zhihui Jia, Zhaokang Yuan, Wei Yang

**Affiliations:** 1Jiangxi Province Key Laboratory of Preventive Medicine, School of Public Health, Nanchang University, Nanchang 330006, Jiangxi, China; 2School of Community Science, University of Nevada, Reno, NV 89557, USA

**Keywords:** high school students, sexual identity, adverse childhood experiences, non-suicidal self-injury, less developed areas of China

## Abstract

*Purpose*: The objective of this study was to examine the influence of sexual identity and adverse childhood experiences (ACEs) on non-suicidal self-injury (NSSI) among rural high school students in less developed areas of China. *Methods*: Behavior risk factors data collected from 1810 students from a high school in Jiangxi province, China. Five measures of childhood abuse and household dysfunction were summarized, and ACE was divided into 0, 1, 2, 3–5 ACEs. Logistic regression analysis was used to explore the influence of sexual identity, adverse childhood experiences, and their interaction with non-suicidal self-injury. *Results*: Compared with heterosexual students, high school students who identify as lesbian, gay, or bisexual (LGB) have a higher tendency of non-suicidal self-injury (AOR = 3.250, 95% CI = 1.69–6.28, *p* < 0.01). There was also a graded relationship between cumulative ACEs exposure and non-suicidal self-injury behaviors (AOR = 1.627, 95% CI = 1.02–2.60, *p* < 0.05). Odds for NSSI are higher among students with both experienced ACEs and identified as LGB (AOR = 2.821, 95% CI = 1.51–5.29, *p* < 0.05). *Conclusions*: Non-suicidal self-injury is associated with ACEs exposure and with those who identify as LGB, and the NSSI odds are greater when students identify as LGB and have experienced ACEs. More interventions to reduce non-suicidal self-injury should focus on LGB and ACEs and more attention needs to be paid to those who identify as LGB and have been exposed to ACEs.

## 1. Introduction

Non-suicidal self-injury (NSSI) is the intentional destruction of body tissue without suicidal intent, using methods that are not socially sanctioned. Common methods of self-injury include cutting, burning, inserting objects beneath the skin, hitting, and biting [1]. Although it is not fatal or has a low fatality rate, it poses a serious public health risk to adolescents [2]. More than 70% of the adolescents who experienced NSSI were having suicidal ideation [3], and among those teenagers with repetitive NSSI, the probability of suicide is 10 times higher than that of teenagers without NSSI [4]. Various studies from Asia [5], Australia [6], Canada [7], Europe [8] and the United States [9] found that the lifetime prevalence rate of adolescent students’ NSSI was between 10% and 32%. By 2020, it is estimated that 15–30 million teenagers will participate in NSSI behaviors [10]. According to the results of a survey of Chinese youth age 10 to 20, the incidence of NSSI behavior was 26.1% in the past 12 months. However, the proportion of rural left-behind teenagers who report at least one NSSI behavior was 47.3%, which was much higher than that of non-left-behind teenagers, and much higher than the average level of the above foreign surveys. Moreover, studies found that the overall incidence of NSSI behavior in this group was high, the duration was long, and the occurrence trend was repeated [11]. 

Several studies have shown that lesbian, gay, or bisexual (LGB) sexual identity are risk factors for adolescent NSSI behavior. There is still a significant correlation between people who identify as LGB and non-suicidal self-injury after controlling for known risk factors such as age, gender, and depression [12]. A recent study found that 35% of the LGB population had participated in NSSI behaviors, 2–8 times more than the heterosexual group. This group’s NSSI thoughts and behaviors were more frequent and lasted longer than the general population [13]. Data from US community surveys suggested an association between sexual identity and exposure to adverse childhood experiences (ACEs). Compared with heterosexuals, sexual minorities (gay/lesbian and bisexual respondents) had a higher rate of ACEs and higher odds of experiencing ACEs. Childhood sexual abuse and a risky family environment, including witnessing parental violence, the relationship strain between the respondent and one or both parents, or living with a problem drinker in the household, were significantly associated with NSSI, along with identifying as sexual minorities in the community sample [14]. Another study also showed that LGB respondents had higher odds of exposure to childhood abuse (physical or sexual) than heterosexuals [15]. Therefore, it is common for LGB teenagers to suffer from strong pressure caused by their families and to have adverse childhood experiences, which may also lead to NSSI [16]. Survey data showed that six out of 10 people in the general population have experienced at least one adverse childhood experience, and 8.7% have reported five or more adverse childhood experiences [17]. Previous studies have shown that there is an association between adverse childhood experiences and sexual identity. Compared with heterosexuals, LGB groups have higher incidence of adverse childhood experiences and higher probability of multiple experiences [15]. Moreover, in the whole life cycle of this group, there is a relationship between adverse childhood experiences and NSSI [18,19]. Adolescents exposed to adverse childhood experiences, such as parental addiction, mental health problems, or witnessing domestic violence have an increased risk of NSSI [20]. At present, there is no research on the interaction between adverse childhood experiences, sexual identity and the influence on NSSI.

As an economically underdeveloped region in China, Jiangxi province has a large population mobility, and a large number of surplus rural labor force enter the city for work and employment. This leads to a lack of education among family members for most rural teenagers [21]. Contrarily, rural high school students who live in schools are a relatively special group of young people. Facing the college entrance examination, they are under great pressure, and the emotionally rich experience caused by the changes in the brain’s own structure during the development of adolescents [22]. They are prone to be troubled by sexual identity selection, more likely to experience negative exposure from family during childhood, and have encountered psychological problems such as introversion, irascibility, and NSSI [23]. Previous studies exploring the role of adverse childhood experiences in the relationship between sexual identity and NSSI mostly relied on adult samples [15,24], and cumulative adverse childhood exposure to NSSI has rarely been studied. In addition, the analysis of the mediating effect was mostly adopted to explore the relationship between the three factors [25]. Based on the above conditions, this study examined 1810 rural high school students in Jiangxi province, an underdeveloped region of China. These samples were set to explore the independent and mutual influence of sexual identity and accumulated adverse childhood experiences on NSSI.

## 2. Methods

### 2.1. Participants and Procedures

In May 2016, a high school in Jiangxi province was selected by the convenient sampling method, and the Youth Risk Behavior Surveillance System (YRBSS) of the Center for Disease Control and Prevention (CDC) was used to design a “Questionnaire on adolescent health risk behaviors in less developed areas of China”. After repeated modification by experts, the Cronbach coefficient of the questionnaire was 0.76. Questionnaires were distributed to all 10th and 11th grade students during a self-study night by study investigators. Students were informed that their participation in the study was voluntary and their responses were anonymous. The self-administered questionnaire took approximately 40 minutes to complete. SAS 9.4 software (SAS Institute, Cary, NC, USA) was used to conduct logistic screening, clear and delete invalid records (Firstly, the scale was then assessed and if a participant had not completed more than 30% of a measure they were classified as incomplete and their data were omitted from the analysis of the scale. Secondly, if they had not completed the ACEs measure, sexual orientation measure or non-suicidal self-injury items, participants were excluded from the main analysis), and 1810 valid questionnaires were collected, with an effective rate of 95.26%. A total of 805 female students (44.48%) and 1005 male students (55.52%) were included. The age distribution was as follows: 96 (5.30%) students ≤ 15 years old, 553 (30.55%) students 16 years old, 867 (47.90) students 17 years old, 294 (16.25%) student ≥ 18 years old.

### 2.2. Measures

**Sexual identity.** Sexual identity was defined by asking students the following question, “Which of the following best describes you?” Responses included heterosexual, gay or lesbian, bisexual, and questioning (the participants were questioning about their sexual identity). Three comparison groups were used for the analyses: (1) LGB; (2) questioning; (3) heterosexual.

**Adverse childhood experiences.** Behavioral Risk Factor Surveillance System (BRFSS) ACE module [15,26] and Kristen Clements-Nolle’sen [27] investigation on the ACE model of American high school students were used for the definition of this variable. The “Questionnaire on adolescent health risk behaviors in less developed areas of China”, defines ACEs into five categories as follows: (1) lifetime sexual abuse—“Have you ever been physically forced to have sexual intercourse when you did not want to?”; (2) physical abuse by an adult—“Have you ever been hit, beaten, kicked, or physically hurt in any way by an adult? (Do not include being spanked for bad behavior)”; (3) household domestic violence—“Have you ever seen or heard adults in your home slap, hit, kick, punch, or beat each other up?”; (4) household mental illness—“Have you ever lived with someone who was depressed, mentally ill, or suicidal?”; and (5) household substance abuse—“Have you ever lived with someone who was a problem drinker or alcoholic or abused street or prescription drugs?” Responses to all ACE questions were dichotomized as Yes versus No. 

Two household dysfunction questions (household mental illness and household substance use) included a response of “don’t know”. Consistent with previous research among American high students using the ACE module, “don’t know” responses were coded as missing for these questions [27]. The five ACE questions were summed to create a total ACE score (range 0–5). The ACE score was further categorized as 0, 1, 2, and 3–5 ACEs [28].

**Non-suicidal self-injury.** Non-suicidal self-injury was assessed with a standardized question from the CDC YRBS, “During the past 12 months, did you ever seriously attempt NSSI?” Responses were dichotomized as Yes versus No.

**Covariates.** We included 15 potential confounding variables from the standardized YRBS, reference to Siobhan O’neill [29] and domestic Li Jianni’s [30] research, it can be divided into the following four aspects: personal factors, family factors, school factors and social factors. 

**Personal factors** included: sex, age, science-humanities for branch of intention. Branch of intention was assessed by asking students: “In your future high school study, which direction will you choose as your study direction, science or humanities? Science includes math, Chinese, English, physics, biology, chemistry; Humanities include math, Chinese, English, politics, history, geography”;

**Family factors** included: parent migrant workers, living with parents, parents often check their whereabouts when they go out, all of the variables were dichotomized as Yes versus No;

**School factors** included: Students who live in school, bullied at school (past year), fight at school (past year), academic pressure. Academic pressure was assessed by asking students: “What level of stress do you feel as a result of your studies?” Ordinal response items were dichotomized as high academic pressure (very stressed, above average stress) versus median/low academic pressure (average stress, below average stress, no stress).

**Social factors** included: current smoking (past 30 days), current alcohol use (past 30 days), in the process of going out, were hurt by others (past year), sexual intercourse (lifetime). In the process of going out, was hurt by others was assessed by asking students, “During the past year, have you ever been physically hurt on purpose in any way by someone who you were dating or going out with?”. Sexual intercourse was assessed by asking students: “Have you ever had sexual intercourse?” all of the variables were dichotomized as Yes versus No.

### 2.3. Data Analysis

Pearson chi-square test was used to compare the unadjusted relationship between personal factors, family factors, school factors, social factors, sexual identity, adverse childhood experiences, and non-suicidal self-injury in rural high school students of China. Binary logistic regression analysis was used to explore the relationship between sexual identity (LGB, questioning, and heterosexual), cumulative adverse childhood experiences (0, 1, 2, 3–5 ACES), and non-suicidal self-injury. Meanwhile, covariates were controlled for, and the adjusted odds ratio (AOR) and 95% confidence interval (95% CI) were calculated. 

Sexual identity × ACE were divided into four groups: (1) heterosexual/0-ACE (reference), (2) LGBQ, (3) ACE, (4) LGBQ/ACE. Multivariate unconditional Logistic regression model was included, and these two groups of objects were combined for interactive analysis. Meanwhile, covariables were controlled, and adjusted odds ratio (AOR) and 95% confidence interval (95% CI) were calculated with a = 0.05 (double side). We entered the data using EpiData 3.0 (The EpiData Association, Odense, Denmark), imported the database into Excel (Microsoft Corporation, Redmond, WA, USA), and then transferred it into SAS version 9.4 (SAS Institute, Cary, NC, USA) for statistical analysis. GraphPad Prism 5.0 (GraphPad Software, San Diego, CA, USA) was used for illustration.

### 2.4. Ethical Statement

All subjects gave their informed consent for inclusion before they participated in the study. The study was conducted in accordance with the Declaration of Helsinki, and the protocol was approved by the Ethics Committee of the second affiliated hospital of Nanchang University. 

## 3. Results

Among the 1810 participants, a total of 123 had detected NSSI behaviors, with a detection rate of 6.80%. In terms of school factors, the NSSI detection rates of high school students with high academic pressure (7.92%) were higher than those with medium/low academic pressure (5.41%), and the NSSI detection rates of high school students involved with fights and bullying at school behaviors were 17.07% and 18.90%, respectively, which were statistically different from those without such behaviors (*p* < 0.05). In terms of social factors, NSSI detection rates of high school students who smoke, use alcohol, in the process of going out was hurt by others, and have sex intercourse are 18.05%, 20.16%, 24.59%, 23.29%, respectively. Compared with those who do not have such behaviors, the differences are statistically significant (all *p* values are <0.01). In terms of individual and family factors, there was no significant statistical difference in the detection rate of NSSI demonstrated in Table 1.

Table 2 shows the unadjusted association between adverse childhood experience, sexual identity and non-suicidal self-injury. In terms of sexual abuse, physical abuse, witnessing household domestic violence, household mental illness history and household substance use/abuse history, NSSI detection rates of high school students were 25.00%, 17.69%, 11.76%, 12.61% and 14.71%, respectively. The differences were statistically significant (all *p* values were <0.01). The NSSI detection rates of high school students with cumulative scores of 0, 1, 2, and 3–5 were 4.17%, 6.82%, 17.98%, and 21.67%, respectively. With the accumulation of ACE scores, high school students were more likely to have NSSI, showing significant statistical differences (*p* < 0.01). In addition, NSSI detection rates of high school students with LGB, questioning and heterosexual identity were 21.21%, 6.56% and 6.20%, respectively, with statistically significant differences (*p* < 0.01).

Utilizing NSSI as the dependent variable, personal factors, family factors, school factors, social factors, sexual identity, ACEs respectively as the independent variables, the binary logistic regression model was included, and standard a = 0.05 was included. The results showed that among the individual factors, females (AOR = 1.592; 95% CI = 1.021–2.482) compared with males, had a higher tendency of NSSI (*p* < 0.05). Being bullied at school (AOR = 2.286; 95% CI = 1.352–3.866) and participation in school fights (AOR = 1.817; 95% CI = 1.039–3.177) had a higher tendency of NSSI than those without such behavior (*p* < 0.05). Among social factors, smoking (AOR = 2.073; 95% CI = 1.092–3.938), alcohol use (AOR = 2.431; 95% CI = 1.372–4.305) high school students with these behaviors have a higher tendency of NSSI than those without behaviors (*p* < 0.05). After controlling for these covariates, sexual identity among LGB students (AOR = 3.220; 95% CI = 2.100–4.935) had a higher tendency of NSSI than those who identify as heterosexual (*p* < 0.001). With the increase of ACE scores, the incidence of NSSI increased (*p* < 0.05): 1 ACE (AOR = 1.63, 95% CI = 1.02–2.60), 2 ACEs (AOR = 4.89, 95% CI = 3.01–6.96) and 3–5 ACEs (AOR = 5.48, 95% CI = 2.72–8.51). See Table 3 and Table 4 for details.

In the binary logistic regression model of interaction (sexual identity × ACE), whether to participate NSSI behavior was the dependent variable, as well as personal factors, family factors, school factors, social factors, sexual identity and sexual identity × ACE (heterosexual/0-ACE, LGBQ, ACE, LGBQ/ACE,) as the independent variables.

The binary logistic regression model was included, and standard a = 0.05 was included. Heterosexual/0-ACE as the control group, AORs and 95% CIs were plotted. After controlling for covariables, the results showed that compared with heterosexual/0-ACE students, LGBQ students (AOR = 1.455, 95% CI = 1.01–2.26), ACE (AOR = 1.894, 95% CI = 1.20–3.00), and LGBQ students with ACE (AOR = 2.821, 95% CI = 1.51–5.29) were more likely to engage in NSSI (*p* < 0.05). See Figure 1 for details.

## 4. Discussion

NSSI is closely related to suicide attempts, and NSSI teenagers have a high suicide risk [31]. This study in developing areas of China with rural high school students as the research object, mainly revealed that personal, family, school, and social factors, such as the covariate control, sexual identity and adverse childhood experiences, forged a link between non-suicidal self-injury. This link extends to less developed areas of China, and to rural high school students and related research on sexual identity and the impact of adverse childhood experiences on non-suicidal self-injury behavior should be carried out here. Consistent with previous studies, students who identified themselves as LGBQ [10,32] had a higher risk of NSSI, and there was a strong correlation between adverse childhood experiences and NSSI [33]. In addition, the results of this study also show that there is a significant interaction between sexual identity and adverse childhood experiences. Students who identify themselves as LGBQ with adverse childhood experiences have a higher detection rate of NSSI.

The results of this study showed that school fights and bullying are risk factors for high school students to engage in NSSI, which is consistent with a study on Canadian teens [25]. Due to the fact that parents go out to work and there is negligence in management for rural high school students, there will be a certain degree of defects in the character of them. The brain’s lack of regulation of negative emotions thus induces the occurrence of NSSI and adverse consequences [34]. Among social factors, high school students with smoking and alcohol behaviors are more likely to engage in NSSI. Teenagers have low alcohol tolerance, and under the stimulation of alcohol, the decline of cognitive ability and emotional management may increase the risk of NSSI [35]. Mäkikyrö [36] found that adolescent smoking is associated with increased behavioral risk of self-harm (OR = 3.32) and suicide (OR = 4.33). The central nervous system of smoking adolescents may be impaired in catechol and serotonin function. This often results in more impulsive and aggressive behaviors. 

After controlling for all covariables, sexual identity among LGB high school students and accumulated ACE are risk factors of NSSI. The higher tendency of NSSI will be along with the increase of ACE scores, which is consistent with the research results of Richard and Kaess. [37,38]. Nowadays, although the public is much more tolerant towards sexual minorities, due to the influence of China’s long-standing traditional ideas, most people are not optimistic about the actual attitude towards sexual minorities. Therefore, once the LGB identity is recognized, sexual minorities are more likely to experience harassment, peer bullying and other insults, resulting in interpersonal difficulties, negative effects on mental health, and more prone to NSSI and other adverse behaviors [39]. As a stimulus, ACE will disrupt the development of the nervous system in children, affect the normal function of the brain’s neurotransmitters and hormones. Thus, increasing the incidence of youth risk behavior [40]. Inconsistent with previous studies, our research showed there is no statistical difference in the NSSI behavior of those who are questioning their sexual identity among rural high school students. A potential reason for that is the influence of traditional Confucian values, Chinese people place great importance on being married and procreation to keep the family bloodline. This value has been deeply rooted in the mind of Chinese people and violation of this value contributes to intolerance toward sexual minorities. So, under pressure, even though their sexual identity is in the LGB group, students choose the vague option-questioning. Secondly, due to the fact that rural schools do not have adolescent sex education courses, parents cannot give correct guidance, so students have not been exposed to complete sex education, lack of sexual knowledge, and then the concept of sexual orientation is vague [41].

The results of this study showed that there is a significant interaction between sexual identity and adverse childhood experiences. Compared with heterosexual students without ACE, students who self-identified as LGBQ with ACE have about three times higher probability of NSSI, which further confirms other survey results [42]. The target population for this study was rural high school students in less developed areas of China. On the one hand, due to the long-time separation from their parents, improper inter-generational upbringing and the lack of a role of father or mother in family education, results in the lag of the cognitive, emotional or behavioral development of rural high school students. High school students are in adolescence, and this is the crucial period of physical and mental development so the lag may change an individual’s performance for themselves and to others. The lack of individual emotional regulation skills may result in self-emotion control by means of self-harm [43]. On the other hand, Tseng and Yang [44] found that family support was negatively correlated with non-suicidal NSSI among adolescents, which also confirmed the conclusion of this study. Most of the parents of rural high school students in less developed areas go out for work. They are exposed to adverse experiences during their growth period, their identities are stigmatized by their peers, and under the domination of rural traditional concepts, society, families and schools have higher cultural expectations for them, which increases their life pressure. For rural high school students, there is no economic ability to relieve pressure through other means by themselves. Parents are the closest people in their life. Only by seeking the help of parents and getting support from the family can they alleviate the pressure, but it is sometimes impossible to get the family support for the first time, and they may adopt avoidance or self-tolerance, leading to serious negative emotions and various psychological problems, including relatively serious psychological disorders and psychological diseases [43]. So, when ACE and LGB interact, the NSSI behavior of rural high school students in less developed areas will be intensified.

These findings suggested that China’s less developed regions should be further focused on public health prevention programs on rural youth groups to identify other risk factors such as increasing or decreasing NSSI. Parents play a vital role in the process of adolescent growth, long-term companionship, harmonious family relationship, and positive guidance to their children, all of which may reduce the self-injury behavior of rural high school students [45]. In addition, targeted training of high-quality rural high school psychological counselors can help ensure their understanding ability, so as to improve the identification and intervention of early NSSI behaviors, and provide effective support and care for these high school students with ACE/LGB, so as to reduce the incidence of NSSI [46].

## 5. Conclusions

This study examines rural high school students in less developed areas of China, and the results showed that 6.8% of rural high school students had engaged in NSSI in the past 12 months. After controlling for covariates, the sexual identity of LGB group and the high accumulation of adverse childhood experiences both increased the probability of NSSI. In addition, after the significant interaction between sexual identity and adverse childhood experiences, students who identified themselves as LGBQ with adverse childhood experiences had a higher detection rate of NSSI. It is suggested that public health organizations identifying and targeting ACEs and LGB may provide opportunities to intervene and reduce non-suicidal self-injury.

### Strengths & Limitations

This study took a small number of rural high school students in less developed areas of China as samples to explore the relationship between adverse childhood experiences, sexual identity, and NSSI. These questions include the use of sexual identity (LGB, questioning, and heterosexuality), which Chinese people are shy to answer honestly, focusing on the cumulative exposure of child abuse and family dysfunction, and identifying the important population for the prevention of NSSI among high school students with relatively high accumulation of adverse childhood growth experiences, which can provide the foundation and basis for future research.

The limitations of this study are (1) the subjects of the study are only from one high school in China’s Jiangxi province, as such the results are generalized to other developed cities in China or school districts to be cautious, but as a less developed area of China, this is a typical rural high school, and the sample size is enough to represent and show the experience of identity, adverse childhood experiences, and NSSI situations in a rural high school setting; (2) sexual identity measurement was based on one dimension, namely the self-reported sexual identity because of limited sample size LGB students (*N* = 66) and LGB students with NSSI (*N* = 14). Therefore, this study will be lesbian, gay and bisexual merged into one group of students, to ensure enough ability to observe and heterosexual students in some dimensions; (3) we only used one question to measure non-suicidal self-injury, while this question is a standardized YRBS measure, it is not a validated NSSI measure. Our research is preliminary and needs to be replicated using a more detailed and validated assessment of NSSI. (4) Although five indicators to measure adverse childhood experiences and family dysfunction were included in this study according to the Behavioral Risk Factor Surveillance System (BRFSS) ACE module, some ACE indicators in adult monitoring studies, such as emotional abuse, parental separation/divorce and other marital status, are still lacking [15].

## Figures and Tables

**Figure 1 ijerph-16-03158-f001:**
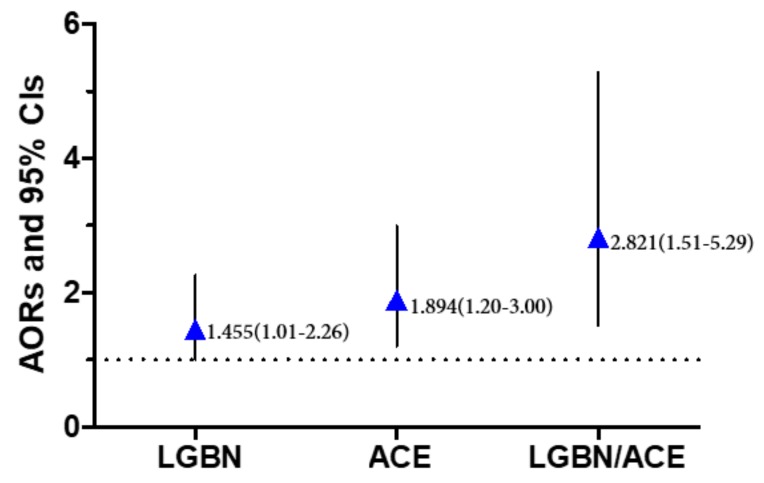
Interacting influence of sexual identity and ACEs on non-suicidal self-injury—2016.

**Table 1 ijerph-16-03158-t001:** Survey results from Students in Jiangxi high school students—2016.

Variables	Options	Total *N*	The Number of NSSI (%) *	χ^2^	*p* Value
**Personal factors**					
Sex				0.02	0.895
	Female	805	54 (6.71)		
	Male	1005	69 (6.87)		
Age (year)				3.74	0.291
	15 or younger	96	11 (11.46)		
	16	553	38 (6.87)		
	17	867	54 (6.23)		
	18 or older	294	20 (6.80)		
Branch of intention				0.00	0.95
	Science	785	53 (6.75)		
	Humanities	1025	70 (6.83)		
**Family factors**					
Parent migrant workers				0.09	0.76
	Yes	874	61 (6.52)		
	No	936	62 (7.09)		
Live with parents				0.04	0.85
	Yes	824	57 (6.92)		
	No	986	66 (6.69)		
Parents often check whereabouts when they go out				0.43	0.51
	Yes	729	53 (7.27)		
	No	1081	70 (6.48)		
**School factors**					
Students who live in school,				1.72	0.19
	Yes	824	49 (5.95)		
	No	986	74 (7.51)		
Academic pressure				4.46	<0.05
	High	997	79 (7.92)		
	Low/Medium	813	44 (5.41)		
Bullied at school (past year)				41.73	<0.001
	Yes	164	31 (18.90)		
	No	1646	92 (5.59)		
Fight at school (past year)				30.08	<0.001
	Yes	164	28 (17.07)		
	No	1646	95 (5.77)		
**Social factors**					
Smoking (past 30 days)				28.68	<0.001
	Yes	133	24 (18.05)		
	No	1677	99 (5.90)		
Alcohol use (past 30 days)				37.58	<0.001
	Yes	124	25 (20.16)		
	No	1686	98 (5.81)		
In the process of going out, was hurt by others (past year)				31.56	<0.001
	Yes	61	15 (24.59)		
	No	1749	108 (6.17)		
Sex intercourse (lifetime)				30.01	<0.001
	Yes	73	17 (23.29)		
	No	1737	106 (6.10)		

* (%) is the detection rate.

**Table 2 ijerph-16-03158-t002:** Adverse childhood experiences (ACEs), sexual identity by non-suicidal self-injury (NSSI) of 1810 high school students surveyed—2016.

Variables	Options	Total *N*	The Number of NSSI (%) *	χ^2^	*p* Value
ACEs					
Sexual abuse				30.24	<0.001
	Yes	56	14 (25.00)		
	No	1754	109 (6.21)		
Physical abuse				26.26	<0.001
	Yes	130	23 (17.69)		
	No	1680	100 (5.96)		
Household domestic violence				27.68	<0.001
	Yes	510	60 (11.76)		
	No	1300	63 (4.84)		
Household mental illness				13.52	<0.001
	Yes	1588	95 (5.98)		
	No	222	28 (12.61)		
Household substance use/abuse				10.68	<0.01
	Yes	102	15 (14.71)		
	No	1708	108 (6.32)		
ACE score				68.09	<0.001
	0	1103	46 (4.17)		
	1	469	32 (6.82)		
	2	178	32 (17.98)		
	3–5	60	13 (21.67)		
Sexual identity				22.52	<0.001
	LGB	66	14 (21.21)		
	Questioning,	244	16 (6.56)		
	Heterosexuality	1500	93 (6.2)		

Note: LGB = lesbian, gay, or bisexual. * (%) is the detection rate.

**Table 3 ijerph-16-03158-t003:** Variable assignment of the potential risk factors among rural high school students.

Factors	Variable Name	Factor Assignment
**Dependent variables**		
NSSI behavior	Y	1 = Yes, 0 = No (Referent)
**Independent variables**		
Sex	X1	1 = Male,0 = Female(Referent)
Age(year)	X2	1 = 16, 2 = 17, 3 = ≥18, 0 = ≤15 (Referent)
Branch of intention	X3	1 = Humanities, 0 = Science (Referent)
Parent migrant workers	X4	1 = Yes, 0 = No(Referent)
Live with parents	X5	0 = No, 1 = Yes(Referent)
Parents often check whereabouts when they go out	X6	0 = No, 1 = Yes(Referent)
Students who live in school	X7	0 = No, 1 = Yes(Referent)
Academic pressure	X8	0 = High, 1 = Low/Medium(Referent)
Bullied at school	X9	1 = Yes, 0 = No(Referent)
Fight at school	X10	1 = Yes, 0 = No(Referent)
Smoking	X11	1 = Yes, 0 = No(Referent)
Alcohol use	X12	1 = Yes, 0 = No(Referent)
In the process of going out, was hurt by others	X13	1 = Yes, 0 = No(Referent)
Sexuality behaviors	X14	1 = Yes, 0 = No(Referent)
Sexual identity	X15	1 = LGB, 2 = Questioning,0 = Heterosexuality(Referent)
ACEs	X16	1 = 1-ACE, 2 = 2-ACEs, 3 = 3–5 ACEs,0 = 0-ACE (Referent)

**Table 4 ijerph-16-03158-t004:** Influence of sexual identity and ACEs on non-suicidal self-injury—2016.

Variable	AOR ^a^ (95% CI)	*p* Value
Sexual identity		
Heterosexual(ref)	1.00	
LGB	3.250 (1.685, 6.282)	<0.001
Questioning	1.042 (0.596, 1.821)	>0.050
ACE score		
0 (ref)	1.00	
1	1.627 (1.020, 2.596)	<0.050
2	4.893 (3.009, 6.957)	<0.001
3–5	5.475 (2.723, 8.510)	<0.001

Notes: AOR = adjusted odds ratio; CI = confidence interval; LGB = lesbian, gay, or bisexual; ref = reference. ^a^ Adjusted for sex, age, branch of intention, parent migrant workers, live with parents, parents often check whereabouts when they go out, students who live in school, academic pressure, bullied at school, fight at school, smoking, alcohol use, in the process of going out was hurt by others, sexuality behaviors.

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
