# Peer review of "Research on Relationships between Sexual Identity, Adverse Childhood Experiences and Non-Suicidal Self-Injury among Rural High School Students in Less Developed Areas of China"

_ijerph, 2019, doi:10.3390/ijerph16173158_

Round 1
Reviewer 1 Report
The purpose of this study was to examine if sexual minorities and those with higher ACES are more likely to engage in NSSI. Unfortunately, in this large school sample, the reports of sexual minority status (n = 66 a point that is not highlighted until the final section of the discussion) and those who engage in NSSI (n = 127) are relatively small (especially given what might be expected in this sample) and so bring to question the validity of this study. There was a limited explanation as to why these factors would be so low.
The authors should consult with an English speaker given there are numerous ways that the text is very confusing and poorly constructed, often due to communication difficulties. I have included a lengthy list here for the authors to pay particular attention to. Other issues with scientific communication though are also confusing and difficult to follow.
There are many terms in the introduction that are poorly defined and lead to a less scientific approach to the review of the literature. Terms include “crowd-based studies.” “incomplete family life,” “concentrated life,” “adolescents… rebellious personality… rich emotions … gradually maturing…”
Abbreviations should be written out e.g., CDC,
What was the rationale for modifications and why is a coefficient of .76 thought to be acceptable. Were there subscales as well?
The description of who completed the the questionnaire was confusing.
Were there more male students in the school?
What was the ACE categorization based on?
It is good to keep the grouping names consistent (school versus campus).
It is not clear that informed consent of parents of minors was obtained. What is the requirement of the Helsinki Declaration for minors?
Editing is needed including spaces added etc.
What do the authors attribute to the low rate of NSSI?
The questionnaire was not well described so when we get to the results they are very confusing (e.g., have had sex behaviors).
The descriptions in Table 1 are sometimes worded poorly and don’t always match up with the descriptions in the results (e.g., students who live on campus – yes vs no).
The “process or going out” is very confusing and unclear.
The multivariate logistic regressions were difficult to understand.
In the discussion, this correlational study often slips into possible causal interpretations that are not justified.
On line 266 the first initials of the authors names should be omitted.
Line 276-8 is confusing.
Overall the discussion is very poorly written and confusing.
What evidence does this study show that family support is important (e.g., line 311).
Change “adopted” to “engaged in” NSSI on line 320.
Line 322 “all” should be changed to “both”.
The conclusion is difficult to follow (especially lines 323 to 328).
Reviewer 2 Report
General Comments:
The authors conducted a study of adverse childhood experiences and sexual identity (and their interaction) associated with non-suicidal self-injury. This is an important topic where the literature is still relatively limited. My one overarching concern is the combining of LGB identities for the analysis. There is a wealth of literature establishing that gay and bisexual individuals differ across several risk factors, and should not be lumped as one single category.
Also, this paper needs to be edited very closely and thoroughly for language and grammar. Additional specific comments are below.
Intro:
Overall, this section (and the discussion) would benefit the most from revising for clarity and grammar.
You describe the associations between sexual identity and NSSI, and between ACE and NSSI; what's missing here is how the ACE experiences qualitatively differ across sexual minority groups. This information would help strengthen the rationale for testing for the interaction between sexual identity and ACE.
What do you mean by "crowd-based studies"?
Methods:
As mentioned above, my biggest concern is that combining LGB individuals into one category will miss several important nuances, and I would strongly recommend examining gay/lesbian and bisexual identities as two individual categories in all analyses.
How was missingness handled? Was any form of imputation used?
Given that this was a cross-sectional study, it would be more optimal to calculate prevalence ratios (such as through log-binomial or Poisson regression) as opposed to odds ratios here.
Did you check the models for multicollinearity?
Results:
Where you use "N" as an abbreviation for "unsure", you can use "Q" as this effectively means unsure (questioning). The "Q" will be much more easily identified by most readers.
The first sentence refers to investigators, not participants: This makes it sound like there were 1,810 investigators for this study, but doesn't specify how many participants there were.
Please add confidence intervals for Figure 1. This is especially important for assessing the significance of the interaction.
Discussion:
An overarching comment: The discussion would benefit greatly from more detail on the forms of discrimination sexual minorities face, and how that can affect NSSI. Some discussion of minority stress theory could be helpful with this.
I am not sure what this sentence means: "Compared with other studies, the survey results from people who identify as heterosexual and uncertainty identity take no statistical difference of all kinds of behavior."
Note the use of a single-item NSSI measure as a limitation.
Some more discussion of social desirability bias would also be helpful.
Round 2
Reviewer 2 Report
The authors addressed most of my comments. The most important recommendation I would add is that in your strengths and limitations, it should be specified that you only had 14 LGB students with NSSI (which is why you couldn't analyze LG and B students separately). This is a very important limitation to explicitly state.
Also, your imputation methods should be added to the paper itself. Otherwise, the paper is fine.
